# Total Phenols from Grape Leaves Counteract Cell Proliferation and Modulate Apoptosis-Related Gene Expression in MCF-7 and HepG2 Human Cancer Cell Lines

**DOI:** 10.3390/molecules24030612

**Published:** 2019-02-10

**Authors:** Selma Ferhi, Sara Santaniello, Sakina Zerizer, Sara Cruciani, Angela Fadda, Daniele Sanna, Antonio Dore, Margherita Maioli, Guy D’hallewin

**Affiliations:** 1Department of Applied Biology, University Larbi Tébessi Tebessa, 12000 Tebessa, Algeria; s_ferhi@yahoo.fr; 2Department of Biomedical Sciences, University of Sassari, Viale San Pietro 43/B, 07100 Sassari, Italy; sara.santaniello@gmail.com (S.S.); sara.cruciani@outlook.com (S.C.); 3Laboratoire d’Obtention de Substances Thérapeutiques (L.O.S.T.), Département de Biologie Animale, Université Des Frères Mentouri-Constantine, 25000 Constantine, Algeria; zerizer.sakina@umc.edu.dz; 4Institute of Sciences of Food Production, National Research Council, Traversa la Crucca, 3. Loc Baldinca Li Punti, 07100 Sassari, Italy; angela.fadda@cnr.it (A.F.); antonio.dore@cnr.it (A.D.); guy.dhallewin@cnr.it (G.D.); 5Institute of Biomolecular Chemistry, National Research Council, Traversa La Crucca, 3. Località Baldinca Li Punti, 07100 Sassari, Italy; Daniele.Sanna@cnr.it; 6Istituto di Ricerca Genetica e Biomedica, Consiglio Nazionale delle Ricerche (CNR), Monserrato, 09042 Cagliari, Italy; 7National Laboratory of Molecular Biology and Stem Cell Engineering-National Institute of Biostructures and Biosystems-Eldor Lab, at Innovation Accelerators, CNR, 40129 Bologna, Italy; 8Center for Developmental Biology and Reprogramming-CEDEBIOR, Department of Biomedical Sciences, University of Sassari, Viale San Pietro 43/B, 07100 Sassari, Italy

**Keywords:** grape leaves, ASE, TP, Antioxidant activities, Antiproliferative, pro-apoptotic effects, Gene expression, Nutraceuticals

## Abstract

Grape leaves influence several biological activities in the cardiovascular system, acting as antioxidants. In this study, we aimed at evaluating the effect of ethanolic and water extracts from grape leaves grown in Algeria, obtained by accelerator solvent extraction (ASE), on cell proliferation. The amount of total phenols was determined using the modified Folin-Ciocalteu method, antioxidant activities were evaluated by the 2,2-diphenyl-l-picrylhydrazyl free radical (DPPH*) method and **^·^**OH radical scavenging using electron paramagnetic resonance (EPR) spectroscopy methods. Cell proliferation of HepG2 hepatocarcinoma, MCF-7 human breast cancer cells and vein human umbilical (HUVEC) cells, as control for normal cell growth, was assessed by 3-(4,5-dimethylthiazol-2-yl)-2,5-diphenyltetrazolium bromide reduction assay (MTT). Apoptosis- related genes were determined by measuring Bax and Bcl-2 mRNA expression levels. Accelerator solvent extractor yield did not show significant difference between the two solvents (ethanol and water) (*p* > 0.05). Total phenolic content of water and ethanolic extracts was 55.41 ± 0.11 and 155.73 ± 1.20 mg of gallic acid equivalents/g of dry weight, respectively. Ethanolic extracts showed larger amounts of total phenols as compared to water extracts and interesting antioxidant activity. HepG2 and MCF-7 cell proliferation decreased with increasing concentration of extracts (0.5, 1, and 2 mg/mL) added to the culture during a period of 1–72 h. In addition, the expression of the pro-apoptotic gene Bax was increased and that of the anti-apoptotic gene Bcl-2 was decreased in a dose-dependent manner, when both MCF-7 and HepG2 cells were cultured with one of the two extracts for 72 h. None of the extracts elicited toxic effects on vein umbilical HUVEC cells, highlighting the high specificity of the antiproliferative effect, targeting only cancer cells. Finally, our results suggested that ASE crude extract from grape leaves represents a source of bioactive compounds such as phenols, with potential antioxidants activity, disclosing a novel antiproliferative effect affecting only HepG2 and MCF-7 tumor cells.

## 1. Introduction

Oxidative stress is a pathogenetic mechanism associated with several diseases, including atherosclerosis, neurodegenerative diseases, such as Alzheimer’s and Parkinson’s disease, cancer, diabetes mellitus, inflammatory diseases, as well as psychological diseases or aging processes [1]. Indeed, increased formation of free radicals (FR) can promote the development of malignancy, and “normal” rates of FR generation may account for the increased risk of cancer development in the elderly [2]. Cancer is the major cause of morbidity and mortality in modern society. The number of deaths by cancer in 2008 was estimated to be 7.6 million, a number predicted to double by 2030 [3]. In developed countries, cancer is the main cause of death after cardiac disease [4]. Many treatments against cancer are possible, such as surgical removal, chemotherapy, radiation therapy and immunotherapy.

Apoptosis, or programmed cell death, is a normal and fundamental event that occurs in a highly regulated and precise manner. This process plays a key role in normal tissue development and maturation, maintaining the homeostasis in the body by controlling the immune system. Apoptosis is the most potent defense against cancer since it is the mechanism used by metazoans to eliminate deleterious cells. Furthermore, a large number of chemo preventive agents exert their effectiveness by inducing apoptosis in transformed cells, as shown both in vitro and in vivo [5,6]. Since apoptosis provides a physiologic mechanism to eliminate abnormal cells, dietary factors affecting apoptosis can elicit an important effect on carcinogenesis. For these reasons, activation of apoptosis by dietary factors in pre-cancerous cells may represent a preventive mechanism (chemoprevention) [6,7].

Nearly 90 out of 121 drugs prescribed to treat cancer originate from plants [8]. The term “nutraceutical” was coined in 1989 by Stephen De Felice to define “food, or parts of a food, that provide medical or health benefits, including the prevention and treatment of disease” [9,10,11]. Many studies demonstrate that grapes are rich in anthocyanins, flavanols, flavonoids, terpenes, organic acids, vitamins, carbohydrates, lipids and enzymes [12,13]. These findings have created considerable interest in grape leaves as a promising source of compounds with nutritional properties and biological potential. Moreover, the use of grape leaves provides a way of solving the disposal problems arising from the large amounts of industrial residues generated by the wine and juice industries [14,15].

Extraction is the most important step to recover and isolate bioactive molecules from plant materials. Various extraction techniques have been developed to obtain nutraceuticals from plants in order to shorten extraction time, reduce solvent consumption, increase extraction yield, improve the quality of extracts and increase pollution prevention [16]. Among those, accelerated solvent extraction (ASE) is a solid-liquid extraction process performed at elevated temperature and under pressure to maintain the solvent in its liquid state. The solvent remains below its critical condition during ASE. The increased temperature accelerates the extraction kinetics and the elevated pressure keeps the solvent in the liquid state, thus achieving a safe and rapid extraction. The only disadvantage of ASE is the high cost of the needed equipment [17].

The aim of the present study was to analyze the polyphenol anti-oxidative and anti-proliferative properties of water and ethanol ASE crude extracts from grape leaves grown in Medea (Algeria).

## 2. Results

### 2.1. Yield and Total Phenolic Content

Table 1 shows the yield and total phenolic content of ethanolic and aqueous ASE crude extracts obtained from grape leaves. The aqueous extract gave a higher total phenolic yield (22.8 ± 3.21%) as compared to ethanol (18.87 ± 0.6%), despite not being statistically significant (*p* = 0.116). However, the ethanolic extracts exhibited larger amounts of TP (around 2.8 times) as compared to the water extract (*p* = 0.001). The ethanol polarity might be responsible for the observed TP content difference.

### 2.2. DPPH and EPR Radical-Scavenging Activity

The antioxidant capability was expressed as the quantity of antioxidant inducing a 50% decrease in DPPH concentration or a 50% inhibition of the hydroxyl radical production (IC50). The quenching efficiency of DPPH or hydroxyl radical is inversely proportional to the IC50. Table 2 shows the IC50 of grape leaves ethanolic and aqueous crude extracts. The ethanolic extract of grape leaves showed higher activity of the scavenging DPPH radical (0.09 mg/mL) as compared to the aqueous extract (0.15 mg/mL) (*p* = 0.035). Ethanolic and water extracts provided IC50 of 0.67 (±0.53) and 0.64 (±0.71) mg/mL respectively. The trapping of hydroxyl radical did not show any significant difference between the two extracts (*p* = 0.181).

### 2.3. Effect of Grape Leaves EACE and WACE Extract on HUVEC Cell Proliferation

Both the ethanolic (EACE) and aqueous (WACE) extracts were not toxic for HUVEC cells, with the IC50 being higher that 2 mg/mL. Ethanolic and water extracts inhibited HUVEC cells proliferation in a dose-dependent manner (*p* = 0.01 and HUVEC cells induced an inhibition of cell growth (96%) at 10 µM (Figure 1)).

### 2.4. EACE and WACE Extract Counteract HepG2 Proliferation

The survival of HepG2 cells was significantly reduced following incubation with ethanol (*p* = 0.001) and water extracts (*p* = 0.001) (cell proliferation is expressed as the mean percentages of viable cells relative to untreated cells) (Figure 2). In addition, inhibition of HepG2 cell proliferation by both extracts were dose-dependent. In particular, IC50 was obtained when 0.7 mg/mL or 1.1 mg/mL of ethanolic or water extracts, respectively, were added to the culture medium. In all cases, ethanolic extracts were significantly more active than water extracts (*p* = 0.001). The maximum growth inhibition was obtained using Cisplatin (93.52%), representing the positive control, followed by 2 mg/mL ethanolic extracts (82.5%) and 2 mg/mL water extracts (68.63%).

### 2.5. EACE and WACE Extracts Influence the Expression of Apoptosis-Related Genes in HepG2 Cells

HepG2 cultured in the presence of EACE or WACE exhibited a significant increase in Bax mRNA levels in a concentration-dependent manner, as compared to untreated control (*p* < 0.05) (Figure 3). Moreover, Bcl-2 gene expression was down-regulated in a concentration-dependent manner (*p* < 0.05) (Figure 4). The effect of ethanolic extracts was more prominent on HepG2 cells than water extracts (*p* = 0.002). In particular, the maximum effect on both Bax and Bcl-2 genes was observed using the highest concentration (2 mg/mL) of ethanolic extracts.

### 2.6. EACE and WACE Extracts Influence MCF-7 Proliferation

Similar to what was observed in HepG2 cells, both crude extracts significantly inhibited MCF-7 cell proliferation (Figure 5). In particular, the IC50 for EACE and WACE of grape leaves was 0.43 mg/mL and 0.71 mg/mL, respectively. The ethanolic extracts were significantly more active than the water extracts (*p* = 0.002). The largest percentage of growth inhibition was obtained by Cisplatin (99.34%), followed by ethanolic (88.56%) and water extracts (79.31%) (Figure 5). Results revealed that MCF-7 cells were more sensitive to extracts than HepG2 cells.

### 2.7. EACE and WACE Extracts Influenced the Expression Of Apoptosis-Related Genes in MCF-7 Cells

Ethanol and water extracts significantly modulated Bax and Bcl-2 mRNA expression levels in MCF-7 cells in a concentration-dependent manner, with Bax expression being significantly upregulated (*p* = 0.001) and Bcl-2 significantly down-regulated (*p* = 0.002). The maximum effect was observed at the highest concentration (2 mg/mL) of ethanolic or water extracts (Figure 6 and Figure 7).

## 3. Discussion

Plant bioactive compounds have drawn increasing attention due to their potent antioxidant properties and their marked effects in the prevention of various oxidative-stress-associated diseases, such as cancer. In the last few years, the identification and development of these compounds or extracts from different plants has become a major area of health- and medical-related research [18]. Phenolic compounds are considered as bioactive compounds, widely present in all parts of plant and crude extracts [19].

In this study, we utilized the accelerator solvent extraction method to prepare crude extracts of grape leaves, grown in Algeria, in ultrapure water and 60% ethanol. We aimed at evaluating the anti-proliferative effects of these extracts on HepG2 hepatocarcinoma cells and MCF-7 breast cancer cells. The amount of total phenols and the antioxidant activity were evaluated by scavenging DPPH* and trapping of hydroxyl radical using EPR-spin trapping technique. Then, cell viability was analyzed by using different concentrations of the extracts.

This study showed for the first time, the extraction of bioactive compounds such as phenolic compounds from grape leaves by ASE. ASE provided fast (10 min), easy (automated technique), safe (no direct exposure to the solvent) and inexpensive (in 34 mL of solvent) extraction, leading to high yields and high phenolic contents. Leelavinothan and Arumugam (2008) found that grape leaves contain 99 mg of gallic acid equivalents (mg GAE)/g of phenolic compounds in 70% hydroalcoholic solvent after 72 h of extraction [20], a value lower than the one obtained with the extraction methods described in the present study and requiring a longer extraction time and more solvent. Orhan et al., (2007) describe a phenolic compound yield of 16,07% by extracting 500 g of *Vitis vinifera* dried powder leaves with 80% ethanol at room temperature (5 L * 6 times) [21]. The pressure exerted by ASE allows the extraction cell to be filled faster and helps to force liquid into the solid matrix. Elevated temperatures enhance the diffusivity of the solvent, resulting in an increased extraction kinetic [22,23,24]. Consequently, ASE may be used to obtain a higher yield in an extremely short time as compared to all previously described methods. Indeed, in recent years, ASE has been successfully applied to the extraction of phenolic compounds from different plant materials, such as grape seeds and skin [25,26,27] apples [28], spinach [29], eggplants [30] and barley flours [31].

Electron paramagnetic resonance (EPR) spin trapping has become an indispensable tool for the specific detection of reactive oxygen free radicals in biological systems [32]. The EPR spin-trapping technique was used to study the ability of ASE grape leaves extracts to quench OH radicals, which are common reactive oxygen species associated with oxidative cell damage [33]. The hydroxyl radical reacts unselectively and very quickly with any chemical compound able to lose a hydrogen atom [34]. Our results indicate that water and ethanol grape leaf extracts possessed similar **^·^**OH radicals quenching activity. In water extract, the content of TP, despite being lower than that of ethanol, was high enough to react with the hydroxyl radicals produced, thus excluding any dose-dependent mechanism in the reaction between antioxidants and **^·^**OH.

DPPH* free radical was used to evaluate the ability of phenolic compounds to transfer labile hydrogen atoms to radicals [32]. Our extracts showed high capability to scavenge DPPH*, due to the presence of different polyphenols, including flavonoids, which can be found in grape leaves [35,36]. Generally, the chemical structure of flavan-3-ol family grants a good antioxidant response towards DPPH. The hydrogen-donating substituents (hydroxyl groups), attached to the aromatic ring structures of flavonoids, allow for a redox reaction able to scavenge free radicals [21,37].

Apoptosis can be activated through two major pathways, the mitochondria-dependent pathway and the death-receptor-dependent pathway. In the mitochondria-dependent signaling pathway, the Bcl-2 family of proteins is divided into two groups: suppressors of apoptosis (e.g., Bcl-2, Bcl-XL, Mcl-1) and activators of apoptosis (e.g., Bax, Bok, Hrk, Bad). The Bax/Bcl-2 ratio might represent a critical factor influencing cell behavior. Suppression of Bcl-2 promotes apoptosis in response to several stimuli, including anticancer drugs [38]. Bax is a pro-apoptotic protein residing in the cytosol in an inactive form and translocating, after activation, to the mitochondria, where it plays an important role in mitochondria-mediated apoptosis. Activated Bax, either in homo-oligomeric form or as complex with other proteins, creates pores in the outer mitochondrial membrane, which leads to the leakage of ions, essential metabolites and cytochrome c from mitochondria to cytosol, thus promoting cell death [39]. Our results demonstrated that grape leaves have an anti-proliferative effect on HepG2 and MCF-7 cells. EACE and WACE markedly inhibited HepG2 and MCF-7 cell viability.

In cells cultured with these extracts, the mRNA levels of the anti-apoptotic factor, Bcl-2, were downregulated, while the expression of the pro-apoptotic gene Bax, was significantly induced. Within this context, other authors have demonstrated that molecules as Diazaphenothiaznes exert an antiproliferative activity in MCF7 cells and C32 human amelanotic melanoma, by regulating BAX and BCL2 gene expression [40,41].

Deepak et al. (2015) show that desert plant extracts are able to induce apoptosis in HepG2 cells. They also describe an upregulation of Bax, Bad, cytochrome c, caspase-3, caspase-7, caspase-9 and poly (adenosine diphosphate-ribose) polymerase [42]. Furthermore, the *Allium atroviolaceum* flower extracts was found to inhibit HepG2 cell growth, revealing a sub-G_0_ cell cycle arrest, changes in morphological features and annexin-V and propidium iodide positive staining, which correlates with Bcl-2 down-regulation and caspase-3 activity [43]. Lu Y et al. (2011) report that injectable seed extracts from *Coix lacryma-jobi* L. induce apoptosis in HepG2 cells, with elevated and prolonged expression of caspase-8, which do not significantly influence the expression of Bcl-2 [44]. Moreover, Jun et al. (2009) report that quercetin can inhibit proliferation and induce apoptosis in HepG2 cells by decreasing the levels of surviving cells and Bcl-2 protein expression, and significantly increasing the protein levels of p53 [38].

We found that the ethanolic crude extracts were able to induce a larger anti-proliferative effect as compared to the aqueous crude extracts, which may be due to the different amount of phenols detected in the two different extracts. Nevertheless, further experiments are needed in order to understand if apoptosis could definitely explain the antiproliferative effects induced by the extracts tested in the present study.

Our extracts showed growth inhibition in MCF-7cells, confirming what has been previously described by other authors using different plant extracts. Blassan et al. (2016) report that *Rubus fairholmianus* root extracts inhibit MCF-7 cells growth via caspase 3/7-induced apoptosis [45]. Reis et al. (2013) report that *Leccinum vulpinum* induces DNA damage, decreases cell proliferation and induces apoptosis in MCF-7 cells [46]. Miris et al. (2011) report that pomegranate (*Punica granatum* L.), at certain concentration, inhibits MCF-7 cell proliferation and induces increased expression of the pro-apoptotic gene Bax and decreased the expression of the anti-apoptotic gene Bcl-2. [47]. ASE extracts of grape leaves grown in Algeria were not cytotoxic for HUVEC cells. Atmaca et al. (2016) report that *Salvia triloba* L. extract has pro-apoptotic and anti-angiogenic effect in prostate cancer cell lines while being not cytotoxic for normal cells [48]. Aghbali et al. (2013) describe the pro-apoptotic potential of grape seeds extracts, confirmed by a significant inhibition of cell growth and viability in a dose- and time-dependent manner without inducing damage to HUVEC non-cancerous cells [49]. Indeed, the bioactive phytochemicals, Honokiol and Magnolol contained in *Magnolia officinalis* and their derivatives show an antiproliferative effect on HepG2 cell proliferation while being unable to elicit any effect on fibroblasts [50].

Finally, the literature strongly suggests that grape is a potential source of antioxidant, anticancer and cancer chemo-preventive phytochemicals. The other parts of the grapes, the skin and seeds, the whole grape by itself, grape-derived raisins and phytochemicals within the grapes have also been found to bear potential anticancer properties in various preclinical and clinical studies [51].

## 4. Materials and Methods

### 4.1. Chemicals and Cells

All solvents used were HPLC (High Performance Liquid Chromatography) grade purified (Merck, Darmstadt, Germany); water was purified using a milli-Qplus system from Millipore (Milford, MA, USA). Reagents employed were of analytical grade; Folin-Ciocalteu reagent and Sodium Carbonate (Na_2_CO_3_) were purchased from Carlo Erba (Milan, Italy); DPPH (2,2-diphenyl-1-picryhydrazyl) and gallic acid (3,4,5-trihydroxybenzoic acid) were purchased from Sigma-Aldrich, 5,5-dimethyl-1-pyrroline-*N*-oxide (DMPO) spin trap and dimethyl sulfoxide (DMSO) were purchased from Sigma-Aldrich (Milan, Italy).

HepG2 and MCF-7 cells were obtained from the Hospital of Cagliari, 09121, Cagliari, Italy. HUVECs cells were obtained from Gibco™ (Grand Island, NY, USA). Cisplatin was obtained from the Oncological Services Hospital of Sassari, Italy.

Dulbecco’s phosphate buffered saline (DPBS) was purchased from Euroclone (Milano, Italy); Dulbecco’s modified Eagle’s Medium with phenol red (DMEM) and fetal bovine serum (FBS) from Life Technologies (Grand Island, NY, USA); Medium 200 and LSGS (5-003-10) from Gibco™. TRIzol reagent, SuperScript^®^ VILO™ cDNA Synthesis Kit, Platinum Quantitative PCR Supermix UDG Kit, SybrGreen I, primer and fluorescein from Life Technologies (Grand Island, NY, USA). L-glutamine, Penicillin, Streptomycin, nonessential amino acids from Euroclone (Milano, Italy). (3-(4,5-dimethylthiazol-2-yl)-2,5-diphenyltetrazolium bromide) tetrazolium reduction MTT Cell Proliferation Assay ATCC^®^ 30-1010K kit was purchased from Invitrogen Co. All the primer sequences are represent in Table 3.

### 4.2. Plant Material

Mature leaves from the *Vitis vinifera* L. apical portion were collected in Medea, Algeria in August. Leaves were rinsed with tap water and dried at room temperature (25 ± 3 °C). Finally, they were ground into a fine powder and kept in the dark at 5 °C in a sterile bag and under vacuum for further use.

### 4.3. Extraction Procedure

ASE was performed on a Dionex ASE 350 (Dionex Thermo FisherScientific Inc., Waltham, MA, USA). Powdered leaves (1 g) were weighed into a 22 mL Dionex (ASE 350) stainless-steel cell. The cells were equipped with a stainless-steel fit and a cellulose filter. The optimized operating conditions for ASE extraction are indicated in Table 4.

Two solvents were tested: ethanol 60% (v/v) and water. The extraction was performed in quadruplicate. After the extraction process, water extracts were immediately freeze-dried whereas the ethanolic ones were first evaporated under a nitrogen flow to remove ethanol, then freeze dried. The freeze-dried extracts were weighed and stored at −80 °C until analysis. Accelerated solvent extraction was performed with the lowest extraction temperature to avoid the maximum degradation of thermolabile compounds.

The extraction yield was calculated as follows:(1)$$\mathrm{Yield}\%=\frac{\left(\mathrm{the}\;\mathrm{weight}\;\mathrm{of}\;\mathrm{freeze}-\mathrm{dried}\;\mathrm{recover}\right)}{1\;\mathrm{gram}\;\left(\mathrm{initial}\;\mathrm{weight}\;\mathrm{of}\;\mathrm{leaf}\;\mathrm{powder}\;\mathrm{used}\right)}\times 100$$

### 4.4. Total Phenolic (TP) Content

The total phenolic content was measured using the modified Folin-Ciocalteu method [52,53,54]. 1 mg of each lyophilized extract was mixed with 9 mL of cold ethanol (80%) (1:10 *w*/*v*), vortexed (Stuart, U.K. model SA8.) at 1600 rpm for 2 min and centrifuged (ALC-Centrifuge 4227R, Milan, Italy) at 16,000× *g* for 15 min at 4 °C. 200 µL of each extract were mixed with the Folin-Ciocalteu reagent (1 mL) and allowed to react for 8 min before adding 800 µL of sodium carbonate solution (0.075 mL^−1^). The mixture was incubated in the dark for one hour at room temperature (20 ± 3 °C) followed by an additional hour at 0 °C. The absorbance was read at 760 nm with a spectrophotometer (8453 Agilent Technologies, Santa Clara, CA, USA).

Results were expressed as milligrams of gallic acid equivalent/g of dry weight on the basis of a gallic acid calibration curve (50 to 500 mg/L with R^2^ = 0.996).

### 4.5. Antioxidant Activity

#### 4.5.1. Spin Trapping Assay of the ^•^OH Radical

The hydroxyl radical scavenging activity was determined with the spin trapping method coupled with electron paramagnetic resonance spectroscopy according to Fadda et al. [34]. The hydroxyl radicals were generated by the Fenton reaction and trapped with a nitrone spin trap 5,5-dimethyl-pyrroline N-oxide (DMPO) [55]. 20 mg of the freeze-dried extract mixed with 1 mL of ultrapure water degassed under nitrogen flow was prepared as stock solution. Serial dilutions were prepared from the stock solution, and depending on the results, the correct concentration for each extract was established. 100 μL of the diluted samples were mixed with Fe(II) sulfate 0.1 mM (100 μL), 112 μL DMPO 26 mM (112 μL) and H_2_O_2_ 1 mM (100 μL).

The DMPO-OH adduct was detected with a Bruker EMX EPR spectrometer operating at the X-band (9.4 GHz) using a Bruker Aqua-X capillary cell. The EPR instrument was set under the following conditions: modulation frequency, 100 kHz; modulation amplitude, 1 G; receiver gain, 1 × 10^5^; microwave power, 20 mW. EPR spectra were recorded at room temperature immediately after the preparation of the reaction mixture. The concentration of the spin adduct DMPO-OH was estimated from the double integration of spectra. The hydroxyl radical scavenging activity was expressed as IC50 on the basis of the percentage of inhibition calculated as follows:(2)$$\%\;\mathrm{inhibition}\;=\frac{\left(A_{0}-A_{s}\right)}{A_{0}}\times 100$$
where A_0_ is the intensity of the spin adducts without extract and A_s_ is the absorbance of the adduct after the reaction with the extract. Different sample’s concentrations were used to calculate the IC50, that is, the extract concentration that halves the concentration of hydroxyl radical adduct of the blank. Three replications were performed for each dilution.

#### 4.5.2. DPPH

The radical scavenging activity of ethanolic and water extracts of grape leaves was determined spectrophotometrically with the DPPH test [56].

30 µL ASE ethanolic and water crude extract at different concentrations (0.05, 0.1, 0.2 mg mL^−1^) were mixed with 3 mL of a DPPH methanol solution (0.3 mM). A blank solution was prepared using methanol instead of the extract.

Solutions were stored in the dark at room temperature for 30 minutes. The absorbance was measured at 518 nm and converted into the percentage of inhibition using the following equation:(3)$$\%\;inhibition=\frac{A_{0}-A_{s}}{A_{0}}\times 100$$
where A_0_ is the absorbance of the sample without extract and A_S_ is the absorbance of the sample after the reaction with the extract. The DPPH radical scavenging activity was expressed as IC50. Three replications were performed for each dilution.

### 4.6. Cell Culture

HepG2 and MCF-7 cells were maintained in Dulbecco’s modified Eagle’s Medium with phenol red (DMEM), supplemented with 10% heat-inactivated fetal bovine serum (FBS), 200 of μM L-glutamine, 200 U/mL of penicillin, 10 μg/mL of streptomycin and 0.1 mM of non-essential amino acids. HUVEC cells were cultured in Medium 200 (Gibco™), containing LSGS (5-003-10; Gibco™), 200 U/mL of penicillin and 10 μg/mL of streptomycin. Cells were grown in 75 cm^2^ tissue culture flasks in the culture incubator at 37 °C with 5% CO_2_ and saturated humidity.

### 4.7. MTT Viability Assay

The anti-proliferative activity of ethanolic and aqueous ASE extracts of *Vitis vinifera* L. leaves on HepG2, MCF-7 and HUVEC cells was determined using a cell viability test.

The MTT [3-(4,5-dimethylthiazol-2-yl)-2,5-diphenyltetrazolium bromide] tetrazolium reduction assay is a colorimetric assay based on the ability of functional mitochondria to reduce by succinate dehydrogenase enzyme an insoluble formazan crystal, which displays a purple color [57].

Then, the effects of the treatments on the overall growth of a particular cell population were assessed by determining the number of living cells remaining in the analyzed cell culture. After counting, HepG2, MCF-7 and HUVEC cells were seeded on a 96-well plate at concentration of 10,000/well in 200 µL and incubated at 37 °C in a 5% CO_2_ incubator (Thermo Fisher Scientific, Waltham, MA, USA).

After 24 h, the medium was replaced with fresh medium containing compounds tested (ethanol and aqueous ASE crude extract) at concentration of 0.5 mg/mL, 1 mg/mL and 2 mg/mL. The negative control is performed in growing medium but positive control is prepared in medium supplemented with cosplatin 10 µM. Every test was repeated three times. After one day, we again substituted medium with or without compounds and repeated the same treatment (treatment 2). The MTT substrate was prepared in a sterile Dulbecco’s phosphate buffered saline (DPBS), then added to cells in culture at a final concentration of 650 µg/mL and incubated for 3 h in the culture incubator at 37 °C with 5% CO_2_ and saturated humidity. After incubation, the medium was removed by aspiration and 200 µL/well Dimethylsufoxide DMSO (Sigma Aldrich) was added to each well. Absorbance was read at 570 nm in a Gemini EMMicroplate Reader (Molecular devices). The percentage of cell proliferation was calculated relative to control wells designated as 100% viable cells using the following formula:(4)$$\frac{\left(\mathrm{At}-\mathrm{Ab}\right)}{\left(Ac-Ab\right)}\times 100=\%\;cell\;proliferation$$
where At = absorbance value of test compound (ASE extract), Ab = absorbance value of blank (medium alone), Ac = absorbance value of control.

### 4.8. Gene Expression

HepG2 and MCF-7 cells were plated into 24-well cell culture plates (60,000 cells/500 μL for each well) in culture medium with ethanolic and aqueous ASE extracts of grape leaves to evaluate the expression levels of apoptotic-related genes. Extracts were prepared fresh just before each experiment and dissolved in DMEM.

After treatment, the total RNA was isolated using TRIzol reagent and quantified by measuring the absorbance at 260/280 nm (NanoDrop 2000, spectrophotometer Thermoscientific ND8008, Thermo Fisher Scientific, Waltham, MA, USA). Approximately 1 µg of total RNA was reverse-transcribed to cDNA by SuperScript^®^ VILO™ cDNA Synthesis Kit (Life Technologies, Grand Island, NY, USA).

Quantitative polymerase chain reaction was run in triplicate using a CFX Thermal Cycler (Bio-Rad, Hercules, CA, USA). 2 µL of cDNA were amplified in 25 µL reactions using Platinum Quantitative PCR Supermix UDG Kit. A Supermix 2X was mixed with Sybr Green I, 0.1 µM of primer and 10 nM fluorescein (Life Technologies, Grand Island, NY, USA). Relative target Ct (the threshold cycle) values of Bcl-2 and Bax were normalized to GAPDH, as housekeeping gene. The mRNA levels of cells treated with ethanolic and aqueous ASE extract were expressed using the 2^−ΔΔCt^ method [58], relative to the mRNA level of the untreated sample for each experiment.

### 4.9. Statistical Analysis

Results are expressed as mean ± standard deviation (SD) and were analyzed by ANOVA with Duncan’s multiple range tests procedure (DMRT) and Student’s t-test using 25.0 SPSS Windows software. Differences were considered significant for *p* < 0.05.

## 5. Conclusions

In the present study, we revealed for the first time that accelerator solvent extraction yielded a higher extraction rate of total phenols and antioxidant activity in an extremely short time. The extract obtained from grape leaves grown in the Medea region (Algeria) exhibited an antiproliferative effect on MCF-7 breast cancer cells and HepG2 hepatocarcinoma cells. Moreover, considering previous reports by other authors and the present results that provide evidence for the modulation of Bax/Bcl2 mRNA levels by leaf extracts, which affects the balance between apoptosis and cell survival, it may be concluded that these extracts could be used as an easily accessible source of natural antioxidants, and as a matrix to prepare drugs counteracting distinctive cancer cells’ proliferation.

## Figures and Tables

**Figure 1 molecules-24-00612-f001:**
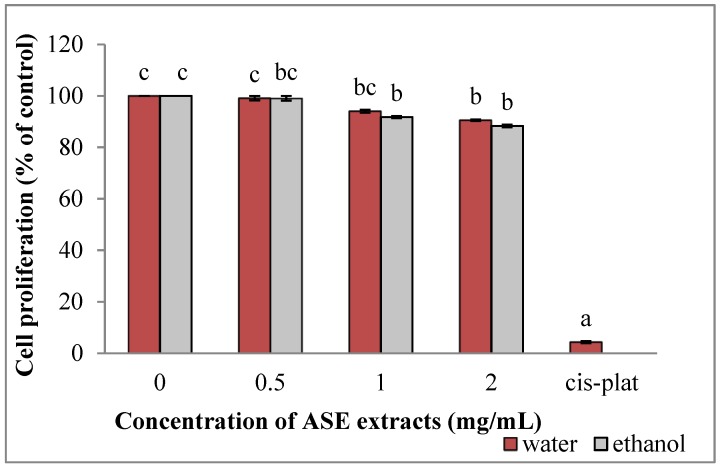
Effect of ASE crude extracts on HUVEC cell proliferation (untreated group: concentration = 0). Data are expressed as mean ± SD, n = 3. Bars marked by unlike letters within a group are significantly different at *p* < 0.05, according to Duncan’s Multiple Range Test (DMRT).

**Figure 2 molecules-24-00612-f002:**
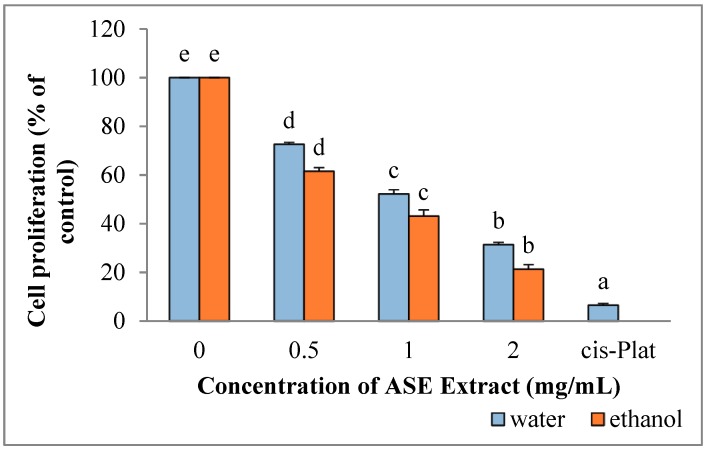
Effect of ASE crude extracts on of HepG2 cell proliferation. Each value is expressed as mean ± SD, n = 3 (untreated group: concentration = 0). Bars marked by unlike letters within a group are significantly different at *p* < 0.05, according to Duncan’s Multiple Range Test (DMRT).

**Figure 3 molecules-24-00612-f003:**
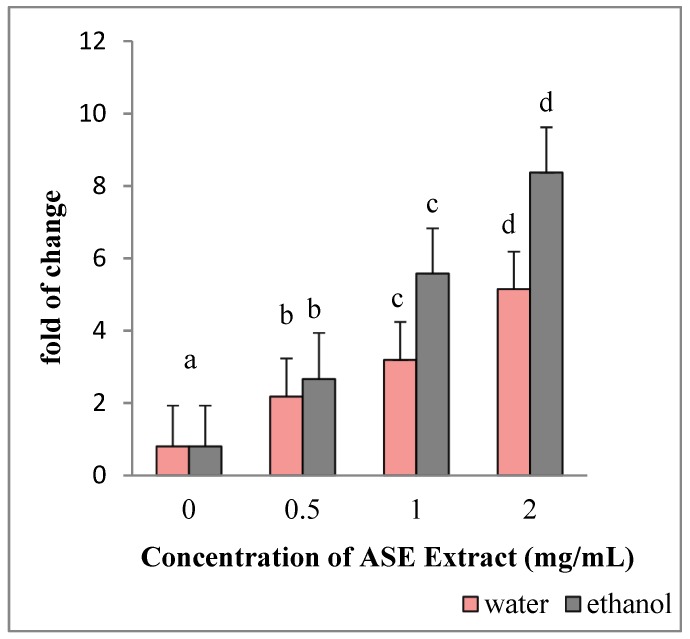
Effect of ASE crude extracts on Bax gene expression in HepG2 cells. The mRNA levels for each gene are expressed as fold of change (2^−∆∆Ct^) relative to the untreated control (defined as 1) (mean ± SD; n = 3) and normalized to the Glyceraldehyde-3-Phosphate-Dehidrogenase (GAPDH). Data are expressed as mean± SD referred to the control. Bars marked by unlike letters within a group are significantly different at *p* < 0.05, according to Duncan’s Multiple Range Test (DMRT).

**Figure 4 molecules-24-00612-f004:**
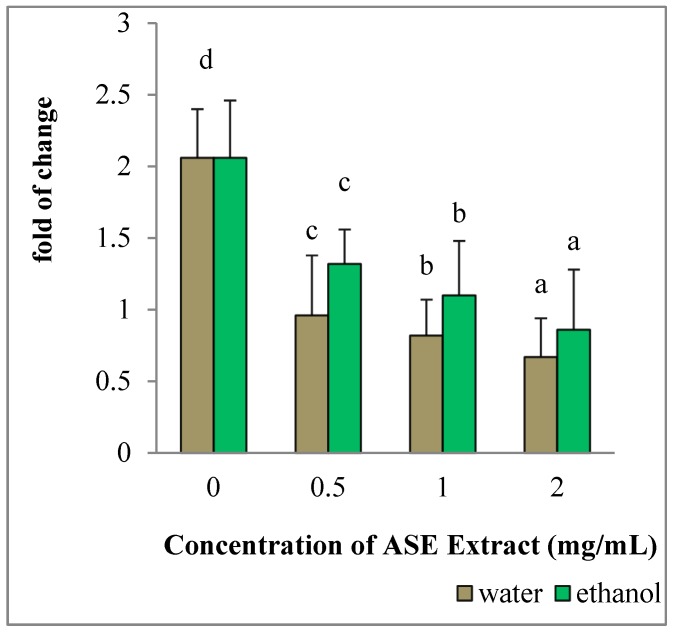
Effect of ASE crude extracts on Bcl-2 gene expression in HepG2 cells. The mRNA levels for each gene are expressed as fold of change (2^−∆∆Ct^) relative to the untreated control (CTRL-), defined as 1 (mean ± SD; n = 3), and normalized to the Glyceraldehyde-3-Phosphate-Dehidrogenase (GAPDH). Data are expressed as mean ± SD referred to the control. Bars marked by unlike letters within a group are significantly different at *p* < 0.05, according to Duncan’s Multiple Range Test (DMRT).

**Figure 5 molecules-24-00612-f005:**
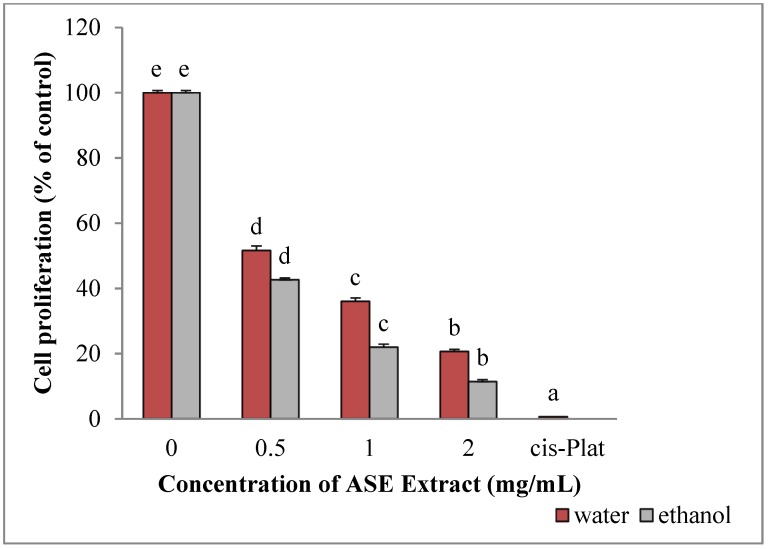
Effects of ASE crude extracts on MCF-7 cell proliferation (concentration 0 corresponding to the untreated group). Data are expressed as mean ± SD, n = 3. Bars marked by unlike letters within a group are significantly different at *p* < 0.05, according to Duncan’s Multiple Range Test (DMRT).

**Figure 6 molecules-24-00612-f006:**
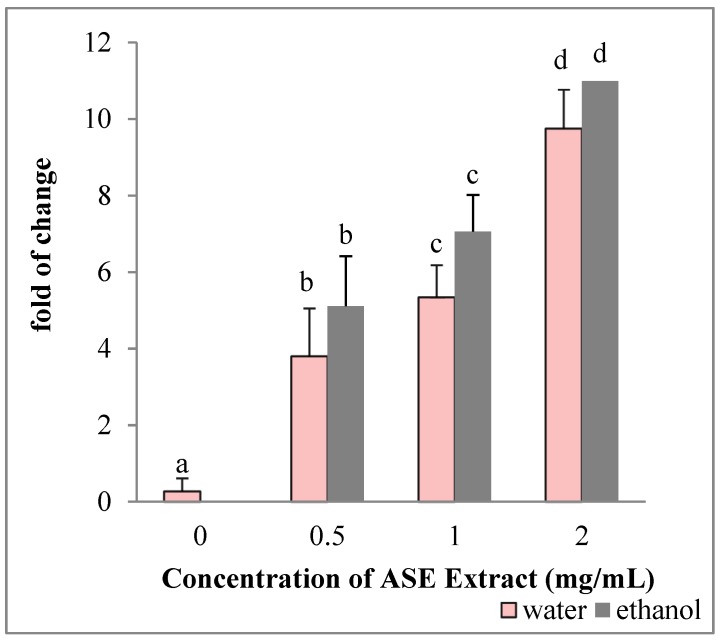
Effect of ASE crude extracts on Bax gene expression in MCF-7 cells. The mRNA levels are expressed as fold of change (2^−∆∆Ct^) as compared to untreated HepG2 cells (defined as 1) (mean ± SD; n = 3) and normalized to GAPDH. Bars marked by unlike letters within a group are significantly different at *p* < 0.05, according to Duncan’s Multiple Range Test (DMRT).

**Figure 7 molecules-24-00612-f007:**
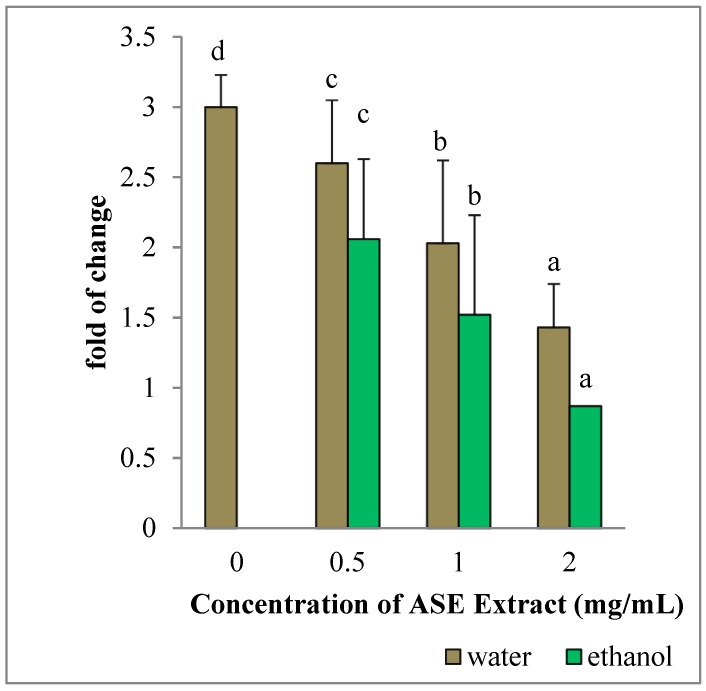
Effect of ASE crude extracts on Bcl-2 gene expression in MCF-7 cells. The mRNA levels are expressed as fold of change (2^−∆∆Ct^) as compared to untreated HepG2 cells (defined as 1) (mean ± SD; n = 6) and normalized to GAPDH. Bars marked by unlike letters within a group are significantly different at *p* < 0.05, according to Duncan’s Multiple Range Test (DMRT).

**Table 1 molecules-24-00612-t001:** Yield extraction (%), total phenols and EPR-spin trapping and DPPH-radical scavenging activity (IC50) of ethanolic and water crude extracts obtained by accelerator solvent extraction (ASE).

Type of Extracts	Total Phenols (mg GAE/gr DW ± SD) ^y^	Yield (% ± SD)	IC50·OH (mg/mL ± SD)	IC50 DPPH (mg/mL ± SD)
**WACE**	55.41 ± 0.11 ^a^	22.8 ± 3.21 ^a^	0.67 ± 0.53 ^a^ R^2^ = 0.9791	0.15 ± 0.41 ^a^ R^2^ = 0.9711
**EACE**	155.73 ± 1.20 ^b^	18.87 ± 0.6 ^a^	0.64 ± 0.71 ^a^ R^2^ = 0.9989	0.09 ± 0.32 ^b^ R^2^ = 09922

^X^ Values within the same column with the unlike letters are not significantly different at *p* ≥ 5; n = 4. ^y^ GAE: gallic acid equivalent; DW: Dry weight; SD: standard deviation; IC50: sample concentration at which 50% of the free radical activity was inhibited. a: ASE water crude extract; b: ASE ethanolic crude extract. The unlike letters represent values significantly different at *p* < 0.05

**Table 2 molecules-24-00612-t002:** IC50* of grape leaves ethanolic and water ASE crude extracts on MCF-7, HepG2 and HUVEC cells.

Extract		MCF-7	HepG2	HUVEC
	Cells			
**WACE ^y^ IC50* (mg/mL)**		0.71	1.1	>>2
**EACE ^x^ IC50* (mg/mL)**		0.43	0.7	>>2

* IC50: sample concentration at which 50% of cell proliferation was inhibited; ^y^ WACE: ASE aqueous crude extract; ^x^ EACE: ASE ethanolic crude extract.

**Table 3 molecules-24-00612-t003:** Primers sequences used for real-time PCR reactions.

Primers	Forward	Reverse
hGAPDH	GAGTCAACGGATTTGGTCGT	GACAAGCTTCCCGTTCTCAG
BAX	TCTGACGGCAACTTCAACTG	TTGAGGAGTCTCACCCAACC
BCL-2	AGGATTGTGGCCTTCTTTGA	ACAGTTCCACAAAGGCATCC

**Table 4 molecules-24-00612-t004:** Conditions of ASE extraction procedure.

Temperature (°C)	40
Pressure (PSI)	1500
Number of Cycle	2
Extraction time of one cycle (min)	5
Concentration of Ethanol (%)	60 Ethanol/40 water
Type of water used	Ultrapure

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
