# Peer review of "Total Phenols from Grape Leaves Counteract Cell Proliferation and Modulate Apoptosis-Related Gene Expression in MCF-7 and HepG2 Human Cancer Cell Lines"

_molecules, 2019, doi:10.3390/molecules24030612_

Round 1

Reviewer 1 Report

I carefully read the manuscript entitled “Total Phenols from grape leaves counteract cell proliferation and induce apoptosis in MCF-7 and HepG2 human cancer cell lines” for possible publication in Molecules.

I would like to emphasize that minor revision is necessary for publication of this manuscript in the Molecules.

Here are some suggestions to improve this manuscript:

English should be checked by Native speaker.

All abbreviations used in the text of manuscript should be      explained.

Introduction section should be more focused on the topic of      manuscript.

Authors should present in the table the IC50 values of      tested extracts against cancer and normal cell lines.

5.     Some references throughout the text should be adjusted to the journal pattern.

Author Response

1. English should be checked by Native speaker.
Authors: We thank the referee for his/her review of our paper and for comments.
The revision of our manuscript has been now checked by an English mother tongue colleague.
2. All abbreviations used in the text of manuscript should be explained.
Authors: All abbreviations are checked and modified by the authors.
3. Introduction section should be more focused on the topic of manuscript.
Authors: The introduction session was modified and improved with more references.
4. Authors should present in the table the IC50 values of tested extracts against cancer and normal cell lines.
Authors: Authors inserted table with IC50 values of cancer and healthy cells (Table 4).
5. Some references throughout the text should be adjusted to the journal pattern.
Authors: Authors adjusted references and added DOI, PMID or URL of each one.

Reviewer 2 Report

The manuscript by Ferhi et al. described the antioxidative and antiproliferative effects of the accelerator solvent extract (ACE) obtained from grape leaves of Vitis vinifera L. grown in Algeria. Comments about this manuscript are listed below:

Major:

This study lacked appropriate control for each activity examined, such as DPPH assay,      EPR spectroscopy assay, and MTT assay.

Authors claimed the proapoptotic effect of the ACE extract on HepG2 and MCF-7      cells simply based upon the mRNA level of BCL2/BAX ratio. This conclusion was TOO PRELIMINARY, as other antiapoptotic BCL-2 family members aside from BCL-2 may compensate the decrease of BCL-2. The authors MUST      perform at least the immunoblotting for the increased levels of cleaved and therefore activated form of caspase 3 or cleaved      PARP which reflects the activation of caspase 3.

The novelty of this manuscript is not obvious. What is the unique finding of this manuscript? Did the ACE extraction lead to higher yield of the antioxidative phenol compoundsIs the species of grape, Vitis vinifera L.,      unique in Algeria? Or others? The authors MUST justify the novelty of      their findings

Minor:

1.          In the Figures, please integrate the data of the two cells lines in a single Figure according to the activity examined.

2.          Some, but not many, sentences in this manuscript were not easy to comprehend. Professional English editing is recommended.

Author Response

Reviewer 2:
Comments to the Author
Major:
1. This study lacked appropriate control for each activity examined, such as DPPH assay, EPR spectroscopy assay, and MTT assay.
Authors: We thank the referee for his/her review of our paper and for comments. Control for each activity:
▪ DPPH and ERP: for the negative control A0, we performed the test without extracts. 3ml of DPPH solution was added to 30 μl of methanol for negative control (A0).
▪ A0 is the Absorbance of the spin adducts without extract (ERP) negative control.
▪ Positive control did not use because we needed to compare the trapping capacity between ethanol and water extract.
▪ MTT negative control is 0 mg/ml concentrations of extract, corresponding to the growing medium.
▪ Positive control was demonstrated by Cis-platinum (chemical anticancer).
2. Authors claimed the proapoptotic effect of the ACE extract on HepG2 and MCF-7 cells simply based upon the mRNA level of BCL2/BAX ratio. This conclusion was TOO PRELIMINARY, as other antiapoptotic BCL-2 family members aside from BCL-2 may compensate the decrease of BCL-2. The authors MUST perform at least the immunoblotting for the increased levels of cleaved and therefore activated form of caspase 3 or cleaved PARP which reflects the activation of caspase 3.
Authors: Apoptosis can be activated through two major pathways, the mitochondria-dependent pathway and the death receptor-dependent pathway. In the mitochondria-dependent signalling pathway, the Bcl-2 family of proteins is divided into two groups: suppressors of apoptosis (e.g., Bcl-2, Bcl-XL, and Mcl-1) and activators of apoptosis (e.g., Bax, Bok, Hrk, and Bad). The Bax / Bcl-2 ratio might represent a critical factor influencing cell behaviour. Suppression of Bcl-2 has been demonstrated to promote apoptosis in response to a number of stimuli, including anticancer drugs. Bax is a pro-apoptotic protein which resides in an inactive form in cytosol and after activation it translocate to the mitochondria, where it plays an important role in mitochondria-mediated apoptosis. Activated Bax either in homo-oligomeric form or as complex with other proteins creates pores in the outer mitochondrial membrane, which leads to the leakage of ions, essential metabolites and cytochrome c from mitochondria to cytosol, thus promoting cell death. So this way is the 2nd major pathway between the two ways previously
cited (upon of the report (BAX/Bcl-2) to stimulate cellular death. Our results together with previously described results by other authors, in which they demonstrated a role of synthesis compounds in regulation of proliferative mechanisms and activation of intracellular apoptosis pathway by gene expression analysis [38, 39], prompt us to hypothesise a pro apoptotic effect of the tested extracts.
3. The novelty of this manuscript is not obvious. What is the unique finding of this manuscript? Did the ACE extraction lead to higher yield of the antioxidative phenol compounds? Is the species of grape, Vitis vinifera L. unique in Algeria? Or others? The authors MUST justify the novelty of their findings
Authors: Conclusions are modified. The novelty of the manuscript is represented by the use ASE enrolled to obtain a higher yield of extraction, total phenols and trapping power in an extremely short time as compared to all previously described methods. Moreover, Vitis vinifera L. is the most abundant species and among the identified in Algeria and this study is the first conduct on it in the world, as revealed in our thesis.
Minor:
1. In the Figures, please integrate the data of the two cells lines in a single Figure according to the activity examined.
Authors: The figures will be too charged and illegible and also difficult to understand because for each activity we have three concentrations plus the negative control plus two types of solvents (water + ethanol) in each figure will contain 4 concentration * 2 extracts * 3 cells + 3 cis test = 27 histograms.
2. Some, but not many, sentences in this manuscript were not easy to comprehend. Professional English editing is recommended.
Authors: The revision of our manuscript is done it by an English mother tongue colleague.

Round 2

Reviewer 2 Report

Thank the authors for doing the best to respond to previous comments. All of the responses are satisfactory except the methodology addressing apoptosis. The authors MUST demonstrate the activation of caspases upon drug treatment, as caspase activation is the hallmark of apoptosis.

Author Response

Reviewer 2, report:

Thank the authors for doing the best to respond to previous comments. All of the responses are satisfactory except the methodology addressing apoptosis. The authors MUST demonstrate the activation of caspases upon drug treatment, as caspase activation is the hallmark of apoptosis.

Authors: We thank the reviewer for his/her precious comments and suggestions. As indicated in the abstract and introduction sessions, the aim of this paper was to demonstrate the antioxidant and antiproliferative effect of the plant extracts (obtained by Accelerator Solvent Extractor-ASE) on HepG2 and MCF7 cells and their influences on the expression of apoptosis-related genes. As suggested by the referee, to demonstrate a real activation of the apoptotic process, it is necessary to analyze the caspase activation pathway. For these reasons, we modified the text and the title of the manuscript, highlighting a modulation of apoptosis related genes rather than the activation of apoptosis itself.